# Investigating Gravitationally Lensed Quasars Observable by Nancy Grace Roman Space Telescope

Lindita Hamolli [1,*,†] , Mimoza Hafizi [1,†] , Francesco De Paolis [2,3,†] and Esmeralda Guliqani [4]

1 Department of Physics, University of Tirana, Boulevard "Zogu I", 1001 Tirana, Albania; mimoza.hafizi@fshn.edu.al
2 Department of Mathematics and Physics "Ennio De Giorgi", University of Salento, Via Arnesano, 73100 Lecce, Italy; francesco.depaolis@le.infn.it
3 INFN, Sezione di Lecce, Via Arnesano, 73100 Lecce, Italy
4 Department of Mathematics and Physics, University "Fan S. Noli", 7001 Korca, Albania; eguliqani@unkorce.edu.al
* Correspondence: lindita.hamolli@fshn.edu.al
† These authors contributed equally to this work.

**Abstract:** In this work, we investigate the possibility of observing quasars, particularly lensed quasars, by the Nancy Grace Roman Space Telescope (Roman). To this aim, based on the capabilities of the Roman Space Telescope and the results from the quasar luminosity function (QLF) in the infrared band of the Spitzer Space Telescope imaging survey, we calculated the number of quasars expected to be in its field of view. In order to estimate the number of lensed quasars, we develop a Monte Carlo simulation to estimate the probability that a quasar is lensed once or more times by foreground galaxies. Using the mass–luminosity distribution function of galaxies and the redshift distributions of galaxies and quasars, we find that 1 per 180 observed quasars will be lensed by foreground galaxies. Further on, adopting a singular isothermal sphere (SIS) model for lens galaxies, we calculate the time delay between lensed images for single and multiple lensing systems and present their distributions. We emphasize that detailed studies of these lensing systems will provide a powerful probe of the physical properties of quasars and may allow testing the mass distribution models of galaxies in addition to being extremely helpful for constraining the cosmological parameters.

**Keywords:** quasar; galaxy; strong lensing





## 1. Introduction

Gravitational lensing refers to the deflection of light rays from a background source due to the presence of a massive object close to the line of sight. In the case of strong lensing, a source (quasar) will appear multiply imaged, magnified, or deformed into an arc-like shape around a lens (foreground galaxy). Since gravitational lensing depends only on the distribution and geometry of foreground matter, it is an ideal tool for mapping the distribution of dark matter and testing the values of cosmological parameters, such as Hubble constant $H_0$ [1].

Today, there is a well-known discrepancy in the reported measurements of $H_0$. From one hand, the Planck telescope finds $H_0 = 67.4 \pm 0.5 \, \text{km s}^{-1} \, \text{Mpc}^{-1}$ by observing the cosmic microwave background (CMB) [2], which is in good agreement with the measurements from galaxy clustering and weak lensing ($H_0 = 67.4 \pm 1.2 \, \text{km s}^{-1} \, \text{Mpc}^{-1}$ [3]). On the other hand, the *SH0ES* team, using the distance ladder method with type *Ia* supernovae and Cepheids, reported a value $H_0 = 74.0 \pm 1.4 \, \text{km s}^{-1} \, \text{Mpc}^{-1}$ [4], which substantially differs from the Planck ($\Lambda CDM$) result. Another possibility for determining $H_0$ is provided by gravitational lensing through the time-delay method. The H0LiCOW collaboration, measuring the time delay of six lensed quasars, found $H_0 = 73.3^{+1.7}_{-1.8} \, \text{km s}^{-1} \, \text{Mpc}^{-1}$, with an uncertainty of only 2.4% [5]. This method is completely independent of both the supernovae and *CMB*

analyses. Different ground- and space-based surveys, such as the Cosmic Lens All Sky Survey (CLASS [6]), the Sloan Digital Sky Survey (SDSS [7]) and Hubble Space Telescope (HST [8]) provide huge amount of photometric and spectroscopic data, which greatly helped to find gravitationally lensed quasars. These systems can be double systems, such as  the first gravitational lens system, Q0957+561, discovered accidentally by [9] in 1979, triple systems or even richer systems. To date, more than 200 lensed quasars are known[1], and this number is expected to grow rapidly by forthcoming surveys, including the Vera Rubin Observatory's Legacy Survey of Space and Time (LSST), Euclid [10] and Nancy Grace Roman Space Telescope [11].

The Roman Space Telescope, formerly known as the Wide Field InfraRed Survey (WFIRST [12]), is a NASA infrared space telescope in development. It is planned to be launched in May 2027, and will be positioned at the second Lagrange point ($L_2$) of the Earth–Sun system. Its primary mission lifetime is set for 5 years, with the possibility of extending it for an additional 5 years.  One of the primary mission objectives is to investigate the nature of dark energy with a variety of methods. This will be achieved through two instruments, the wide field instrument (WFI) and the coronagraph instrument. The WFI field of view is 0.281 square degrees and carries 8 science filters (*F*062, *F*087, *F*106, *F*129, *F*158, *F*184, *F*146, and *F*213) with overlapping band passes spanning 0.48–2.3 μm. The Roman Space Telescope survey is intended to cover 2000 square degrees over a five-year period with NIR imaging and spectroscopy. It has an angular resolution of about 0.11 arcsec.

The goal of the present work is to examine the capability of the Roman Space Telescope to observe quasars and, in particular, quasars that are lensed by foreground galaxies. Quasars are very powerful and bright distant active galactic nuclei (AGNs). Being very far and with extremely high intrinsic luminosity, they are considered to be important instruments for understanding the content and the history of the universe. Currently, the theoretical work is focused on systems lensed by one galaxy, that is, characterized by a single lens plane, but  also the gravitational lensing by two foreground galaxies[2] located at different redshifts and separation is studied by many authors (see [13–15] for details). In these papers, by modeling the lensing galaxies through the singular isothermal sphere (*SIS*) law, the number of images and their position, as well as the formation of the Einstein rings, were analyzed. Moreover, it was found that about 5% of lensed quasars are caused by multiple systems.

## 2. Quasars Observed by Roman Space Telescope

In this section, we examine the possibility of the Roman Space Telescope to observe quasars. Roman WFI will observe in infrared wavelengths (0.48–2.3 μm), and its survey area is planned to be about 2000 deg$^2$. We use the quasar luminosity function (QLF), which is the comoving number density of quasars as a function of their luminosity and redshift, to find the observable number of quasars. They are luminous in almost all accessible bands, but here, we consider the QFL in the IR band, derived by [16]. Based on the Spitzer Space Telescope imaging surveys, a double power-law function of the form

$$\phi(L, z) = \frac{d\phi}{dlogL} = \frac{\phi^*}{\left[\left(\frac{L}{L^*}\right)^{\gamma_1} + \left(\frac{L}{L^*}\right)^{\gamma_2}\right]},$$ (1)

was extracted, where $\phi$ is the comoving space density of the AGN, $\phi^*$ is the characteristic space density (both in units of comoving Mpc$^{-3}$), $L$ is the rest-frame luminosity at 5 μm, and $L^*$ is the break luminosity at 5 μm, both in units of erg s$^{-1}$ Hz$^{-1}$. The evolution in $L^*$ is a cubic expression (see, e.g., [17]):

$$log_{10}L^*(z) = log_{10}L_0^* + k_1\epsilon + k_2\epsilon^2 + k_3\epsilon^3,$$ (2)

where $\epsilon = log_{10}((z+1)/(z+z_{ref}))$, $L_0^*$ is a free parameter in the fit, which corresponds to the break luminosity at redshift zero, $\gamma_1$ is the faint-end slope, $\gamma_2$ is the bright-end slope and $z_{ref}$ is fixed at 2.5. The coefficients $k_1$, $k_2$, $k_3$, $\gamma_1$, $\gamma_2$, $\phi^*$, and $L_0^*$ are given in Table 1 by Lacy et al. [16].

**Table 1.** Expected number of quasars (column 3) observed by Roman Space Telescope in filter F213 for simple expose and deep survey, both for point-like sources and compact galaxies with half-light radius of $0.3''$. In the fourth column is the redshift limit for each imaging sensitivity, while in the fifth column, the number of the lensed quasars are given.

| F213 | $m_{AB}$ | Number of Quasars ($\times 10^6$) | $z_{max}$ | Number of Lensed Quasars |
|---|---|---|---|---|
| 1 h, Point | 26.2 | 18.4 | 33.6 | 102,000 |
| 1 h, $r_{50} = 0.3''$ | 25.2 | 6.3 | 22.1 | 35,000 |
| 55 s , Point | 23.7 | 1.02 | 11.9 | 5660 |
| 55 s, $r_{50} = 0.3''$ | 22.7 | 0.245 | 7.9 | 1360 |

The number of quasars in the redshift range $z_1$–$z_2$ is obviously given by

$$N = \int_{z_1}^{z_2} \int_{L_{min}}^{L_{max}} \phi(L,z) dL dV_c \tag{3}$$

where $dV_c = D_H \frac{(1+z)^2 D_A^2}{E(z)} d\Omega dz$ is the comoving volume element in the solid angle $d\Omega$ and redshift interval $dz$, $D_H = \frac{c}{H_0}$ is the Hubble distance, and $D_A$ is the angular diameter distance at redshift $z$ (see [18] for details).

The Roman Space Telescope is planned to have eight filters, but in our calculation we consider the F213 filter (the band 1.95–2.30 μm with midpoint at 2.125 μm), which is closer to the band for which the QFL parameters are extracted. In Table 1, the AB magnitude limits achieved at $5\sigma$ in 1 h or 55 s of integration, both for point sources and for compact galaxies with half-light radius of $0.3''$, are given[3]. Using the Vega - AB Magnitude Conversion tool, we convert the AB magnitude (e.g., $m_{AB} = 26.2$) to the apparent magnitude ($m = 24.35$) and then find $L_{min} = L_\odot 10^{\left(-20.42 - log \frac{23.04 \times 10^{-24}}{D_L^2}\right)}$, where $D_L$ is the luminosity distance of the source. Assuming $L_{max} = 10^{47}$ erg s$^{-1}$ in Equation (3), we find the number of quasars expected to be observed by the Roman Space Telescope (survey area, 2000 deg$^2$). Our results for different time intervals (1 h or 55 s) are shown in Table 1. Moreover, we give the limit of the redshift $z_{max}$ for each case. As one can see, for a simple exposure, the number of quasars is about 20 times smaller than for deep surveys.

## 3. Review of Strong Lensing

In strong gravitational lensing, the distances to the lens ($D_L$) and source ($D_s$) are very large. By considering the "thin" lens model, where the mass of the lens is represented as a two-dimensional mass sheet lying perpendicular to the line of sight (known as the lens plane), the lens equation is defined as

$$\vec{\beta} = \vec{\theta} - \vec{\alpha}(\vec{\theta}), \tag{4}$$

where $\vec{\beta}$ is the angular position of source, $\vec{\alpha}(\vec{\theta}) = \frac{D_{LS}}{D_S}\vec{\hat{\alpha}}(\vec{\theta})$ is the scaled deflection angle and $\vec{\hat{\alpha}}$ is the deflection angle[4] (see [19] for details). The lens equation is, in general, a non-linear equation and can be characterized by multiple solutions of $\theta$ for a given source position $\vec{\beta}$. When $\vec{\beta} = 0$, the lens and the source are perfectly aligned, and the image is given by the Einstein ring radius $\theta_E$, which results to be

$$\theta_E = \sqrt{\frac{4GM(\theta)}{c^2} \frac{D_{LS}}{D_S D_L}}, \tag{5}$$

where $D_S$, $D_L$, and $D_{LS}$ are the observer–source, observer–lens and lens–source angular diameter distances, respectively, which can be calculated by the expressions [18],

$$D_L = \frac{c}{H_0(1+z_L)} \int_0^{z_L} \frac{dz}{E(z)}; \quad D_S = \frac{c}{H_0(1+z_S)} \int_0^{z_S} \frac{dz}{E(z)}; \tag{6}$$

$$D_{LS} = D_S - \frac{1+z_L}{1+z_S} D_L; \quad with \quad E(z) = \sqrt{\Omega_m(1+z)^3 + \Omega_k(1+z)^2 + \Omega_\Lambda}. \tag{7}$$

Here, $H_0$ is the Hubble constant and $\Omega_m$, $\Omega_k$, and $\Omega_\Lambda$ are the dimensionless density parameters, i.e., the sum of the cold dark matter and baryonic matter, the space curvature, and the dark energy, respectively.

The scaled deflection angle is related to the scaled lens potential by $\vec{\alpha}(\vec{\theta}) = \vec{\nabla}\Psi(\vec{\theta})$, and the lens Equation (4) takes the form

$$(\vec{\theta} - \vec{\beta}) - \vec{\nabla}_\theta \Psi = 0. \tag{8}$$

Another important quantity in gravitational lensing is the scaled arrival time defined as [19]

$$\tau(\vec{\theta}) = \frac{1}{2}(\vec{\theta} - \vec{\beta})^2 - \Psi(\vec{\theta}). \tag{9}$$

Light rays emitted by a distant source will take some time to reach the observer. This amount of time depends on the path followed by the light rays and will be different if the source is gravitationally lensed or not. Light rays are delayed because of two effects. First, a light ray that is bent is subject to a longer path and therefore needs more time to propagate. This gives rise to a geometrical time delay $\Delta t_{geom}$, which is given by

$$\Delta t_{geom} = \frac{1+z_L}{c} \frac{D_L D_S}{D_{LS}} \Big[\frac{1}{2}(\vec{\theta} - \vec{\beta})^2\Big]. \tag{10}$$

Second, when light rays propagate through a gravitational potential, this results, in general relativity, in an induced time-dilation effect known as the Shapiro effect [20]. The gravitational time delay caused by the lens with redshift $z_L$ is given by

$$\Delta t_{grav} = -\frac{1+z_L}{c} \hat{\Psi}(\vec{\xi}) + const. \tag{11}$$

Here, $\vec{\xi}$ is the position where the light rays cross the lens plane, which corresponds to the two-dimensional angular position $\vec{\theta} = \vec{\xi}/D_L$. The total time delay, $\Delta t$, is the sum of both of them i.e.,

$$\Delta t(\vec{\theta}) = \frac{1+z_L}{c} \frac{D_L D_S}{D_{LS}} \Big[\frac{1}{2}(\vec{\theta} - \vec{\beta})^2 - \Psi(\vec{\theta})\Big] + const. \tag{12}$$

In fact, the unlensed source is not observable, so the time delay $\Delta t$ cannot be measured. However, if the source is time variable, an observable quantity is the time delay between images, which, for two images, labeled as (1) and (2), can be written as

$$\Delta t_{1,2} = \frac{1+z_L}{c} \frac{D_L D_S}{D_{LS}} \Big[\frac{1}{2}(\vec{\theta_1} - \vec{\beta})^2 - \frac{1}{2}(\vec{\theta_2} - \vec{\beta})^2 - \Psi(\vec{\theta_1}) + \Psi(\vec{\theta_2})\Big]. \tag{13}$$

The factor outside the square bracket, noted as $\frac{1+z_L}{c} \frac{D_L D_S}{D_{LS}} = \frac{D_{\Delta t}}{c}$ (where $D_{\Delta t}$ is the so called time-delay distance) is inversely proportional to $H_0$ (see Equations (6) and (7)), while the term in the square bracket depends on the lens configuration and its mass distribution. Of course, observations may allow defining only the positions of image angles $(\vec{\theta_1}, \vec{\theta_2})$, whereas the lens potential $\Psi(\vec{\theta})$ and the source position $\vec{\beta}$ must be inferred via lens modeling.

In the subsequent Sections 3.1 and 3.2, we summarize the gravitational lensing principles in both single and multiple lensing systems, considering a singular isothermal sphere

(SIS) model for lensing galaxies. This model corresponds to a distribution of self-gravitating particles, with a Maxwellian velocity distribution at all radii and one-dimensional velocity dispersion $\sigma_{SIS}$. The density distribution is described by [19]

$$\rho(r) = \frac{\sigma_{SIS}^2}{2\pi G r^2},\tag{14}$$

and the mass of the lensing galaxy inside a radius $\theta_E$ from the galactic center in the galactic plane is given by

$$M(\theta_E) = \frac{\pi}{G}\sigma_{SIS}^2 \theta_E D_L.\tag{15}$$

Replacing the value above in the lens Equation (5), one can obtain

$$\theta_E = 4\pi \frac{\sigma_{SIS}^2}{c^2}\frac{D_{LS}}{D_S}.\tag{16}$$

There are some conclusions about the relation between $\sigma_{SIS}$ and the stellar velocity distribution, $\sigma$. We remark that Ofek et al. [21] predicted $\sigma_{SIS}$ that is not exactly equal to the observed stellar velocity dispersion ($\sigma$), which measures the stellar kinematics related to the gravitational potential of the system, including both the stellar and dark matter components. They proposed that $\sigma_{SIS} = f_e\sigma$, with $(0.8)^{1/2} < f_e < (1.2)^{1/2}$. Additionally, Bolton et al. [22] conducted a study involving 53 massive early-type strong gravitational lens galaxies and found that the ratio between the stellar velocity dispersion and the velocity dispersion obtained from lensing, assuming an isothermal halo model, is approximately one. So, our assumption to calculate the Einstein radius from Equation (16), by substituting the value of $\sigma$ for $\sigma_{SIS}$, appears acceptable.

### 3.1. Single Lensing Systems

Adopting the SIS model to describe the lens galaxy in the single lensing systems, one can find that $\alpha(\theta)$ is independent of $\theta$ and is given by the Einstein radius $\theta_E$. So, in terms of the Einstein angle $\theta_E$, one can write $\alpha(\theta) = \theta_E\frac{||\theta||}{\theta}$. Then, the lens Equation (4) takes the form

$$\theta - \beta = \theta_E\frac{||\theta||}{\theta}\tag{17}$$

Multiple images are obtained only if the source lies inside the Einstein ring, i.e., if $\beta < \theta_E$. When this condition is satisfied, the lens equation has two solutions:

$$\theta_1 = \beta + \theta_E, \quad \theta_2 = \beta - \theta_E.\tag{18}$$

The images at $\theta_1$ and $\theta_2$, are on opposite sides of the lens galaxy and together with the source lie on a straight line. The separation between the formed images always remains equal to twice the Einstein ring's radius, $\Delta\theta = 2\theta_E$. Since $\vec{\alpha}(\vec{\theta}) = \vec{\nabla}\Psi(\vec{\theta})$, the scaled lens potential can be written as

$$\Psi(\theta) = \frac{D_{LS}}{D_S}\frac{4\pi\sigma^2}{c^2}||\theta|| = \theta_E||\theta||,\tag{19}$$

and total time delay can be given by

$$\Delta t(\theta) = \frac{1+z_L}{c}\frac{D_L D_S}{D_{LS}}\left[\frac{1}{2}(\vec{\theta}-\vec{\beta})^2 - \theta_E||\theta||\right] + const.\tag{20}$$

For a time-variable source, the time delay between two lensed images is given by

$$\Delta t_{1,2} = \frac{1+z_L}{2c}\frac{D_L D_S}{D_{LS}}(\theta_1^2 - \theta_2^2).\tag{21}$$

*3.2. Multiple Lensing Systems*

In multiple lensing systems, there are two foreground galaxies along the line of sight towards the source. Their redshifts positioned at $z_1 < z_2$ are smaller than the source redshift, $z_s$. Defining as the "optical axis" the line that passes through the mass center of the foreground galaxy, the lens equation can be written as

$$\vec{x_S} = \vec{x_1} - \vec{\alpha_1}(\vec{x_1}) - \beta\vec{\alpha_2}(\vec{x_2}),\qquad(22)$$

where $\vec{x_S}$ is the unlensed angular position of the source. $\vec{\alpha_1}(\vec{x_1})$ and $\vec{\alpha_2}(\vec{x_2})$ are the physical deflections of a ray when it passes through $\vec{x_1}$ and $\vec{x_2}$, respectively. The angular position on plane of the second galaxy is found by $\vec{x_2} = \vec{x_1} - \beta\alpha_1(\vec{x_1})$, and $\beta$ is defined as $\beta \equiv \frac{D_{12}D_3}{D_2 D_{2S}}$[5]. Here, $D_1$, $D_2$, and $D_S$ are the angular diameter distances between the observer and first lens, second lens and source, respectively, while $D_{12}$ and $D_{2S}$ are the angular diameter distances between both galaxies and between the second galaxy and source (see [23] for detail). Considering the SIS model for lenses, Equation (22) indicates that up to six images of the background source can potentially be produced. In multiple lensing systems, the time delay for the light ray following a deflected light path relative to the undeflected path is given by the sum of delays during the two planes,

$$\Delta t = \frac{1+z_1}{c}\frac{D_1 D_2}{D_{12}}\left[\frac{1}{2}(\vec{x_1}-\vec{x_2})^2 - \beta\Psi(\vec{x_1})\right] + \frac{1+z_2}{c}\frac{D_2 D_S}{D_{2S}}\left[\frac{1}{2}(\vec{x_2}-\vec{x_S})^2 - \Psi(\vec{x_2})\right]\quad(23)$$

where $z_1$ and $z_2$ are, respectively, the redshifts of the foreground and background lenses [24].

## 4. Quasar Lensing with Roman Space Telescope

In this section, we investigate lensed quasars by foreground galaxies. These events are very important for investigations in cosmology. Firstly, we describe an algorithm to estimate the number of lensed quasars that would be observed by the Roman Space Telescope and then give our results.

*4.1. Simulations*

In order to calculate the probability to detect the lensed quasars, we use the Monte Carlo method (see [25] for a more detailed description of our algorithm). For each event, we generate the redshift of the quasar [26] and the redshift of the galaxy [27], provided that it is smaller than the quasar redshift, extract the galaxy mass using the stellar mass function [28], and find its velocity dispersion from the relation between the galaxy's stellar mass and stellar velocity distribution [29]. Using the SIS model for galaxies, we define the Einstein angle $\theta_E$ for each quasar/galaxy pair. For a single galaxy, the probability to reside inside the Einstein angle about the observer–quasar direction would scale as $\theta_E^2/4$. Since [27] give the redshift distribution of 7000 galaxies, we normalize the probability considering the whole number of galaxies, 200 billions (0.25 galaxies per arcsec$^2$, [30]), and find to be $10^8\theta_E^2/14$. We compare this result with a number $n$, uniformly distributed in the interval $(0,1)$, which is extracted by the Monte Carlo code. We keep this pair when its probability is smaller; otherwise, we reject it. The procedure is repeated for all expected quasars to be observed by the Roman Space Telescope. For each aligned system, we calculate the positions of the images and then the time delay between them. In order to estimate the probability that a quasar is lensed by one or more foreground galaxies, we compare the Einstein angle of the system with the angular accuracy of the considered telescope.

Let us give some more details about the distributions mentioned above:

– The *quasar redshift distribution* is taken by Ref. [26], where the authors determined the redshift distribution of 46,420 quasars in the SDSS Data Catalog (Third Data Release), covering an area of 4188 deg$^2$. Most quasars have $z < 2$, with a median value of 1.47 and a mode of 1.85. There are 520 quasars at $z > 4$, including 17 at $z > 5$, and 69 with $z < 0.15$.

– The *galaxy redshift distribution* was found by [27]. The authors analyzed a sample of 7000 distant galaxies observed near the South Galactic Pole by the FORS Deep Field (ESO

VLT), covering a sky area of about $7 \times 7$ arcmin$^2$. This distribution was also confirmed by the analysis of Davison et al. [28].

– The *galaxy mass distribution* obtained by Davidzon et al. [28] is used to extract the mass of the lensing galaxies. They utilized the COSMOS2015 catalog to provide a comprehensive view of the galaxy stellar mass assembly in the redshift range between $z = 0.1$ and 6. Measurements were fitted with a double Schechter function up to $z = 3$ and a single Schechter function beyond that, given by

$$\Phi(M)dM = \left[\Phi_1^*\left(\frac{M}{M_*}\right)^{\alpha_1} + \Phi_2^*\left(\frac{M}{M_*}\right)^{\alpha_2}\right] \exp\left(-\frac{M}{M_*}\right)\frac{dM}{M_*}. \tag{24}$$

Schechter parameters ($M_*, \alpha_1, \Phi_1^*, \alpha_2, \Phi_2^*$) of the COSMOS2015 galaxy stellar mass function (SMF) are found in Table 1 of Ref. [28].

– The *stellar velocity dispersion* $\sigma$ is found by the relation proposed by Zahid et al. [29]. Analyzing data from the SDSS and SHELS, they established the relation between the central stellar velocity dispersion $\sigma$ and the stellar mass of galaxies, given by

$$\begin{cases} \sigma(M) = & \sigma_b\left(\frac{M}{M_b}\right)^{\alpha_1} \quad for \quad M \leq M_b \\ \\ \sigma(M) = & \sigma_b\left(\frac{M}{M_b}\right)^{\alpha_2} \quad for \quad M > M_b. \end{cases} \tag{25}$$

The fit parameters are $log(M_b/M_\odot) = 10.26$, $log(\sigma_b) = 2.073$, $\alpha_1 = 0.403$ and $\alpha_2 = 0.293$. The two indices $\alpha_1$ and $\alpha_2$ define the power law below and above the break point, respectively.

In the next section, we present our results considering observations with the Roman Space Telescope. In our calculations, we use the following cosmological parameters: $\Omega_m = 0.30$, $\Omega_k = 0$, $\Omega_\Lambda = 0.70$ and $H_0 = 70$ km s$^{-1}$ Mpc$^{-1}$.

*4.2. Results*

We generate a sample of 101,000[6] events adopting the redshift distributions of both quasars and galaxies, the galactic mass distribution and stellar velocity dispersion presented in Section 4.1. Assuming the SIS model for the galaxies, we find the probability that a quasar is lensed by one or more foreground galaxies. Considering the Roman Space Telescope, with an angular resolution of 0.11 *arcsec*, we find that about one quasar in 180 in the field of view of Roman is expected to be lensed by foreground galaxies. In the fifth column of Table 1, we give the number of lensed quasars expected to be observed by the Roman Space Telescope for each imaging sensitivity.

Furthermore, our analysis reveals that approximately 10% of configurations can be attributed to the presence of two or more foreground galaxies, and that the probability of a quasar being lensed by more than two galaxies is very low as anticipated.

In Figure 1, the redshift distribution of galaxies and quasars in the case of the single lensing systems is plotted. In these events, two images are formed at the two sides of the quasar. As one can note, the redshift of the galaxy is up to two, while the redshift of the quasars goes up to five. In Figure 2, we present the redshift distributions in the case of a quasar lensed by two galaxies, that is, in the multiple lensing systems that are expected to be observed by the Roman Space Telescope. As can be seen, the redshift of the first galaxy is up to 1.5, and the redshift of the second galaxy is up to 4, while the quasar redshift goes up to 5.

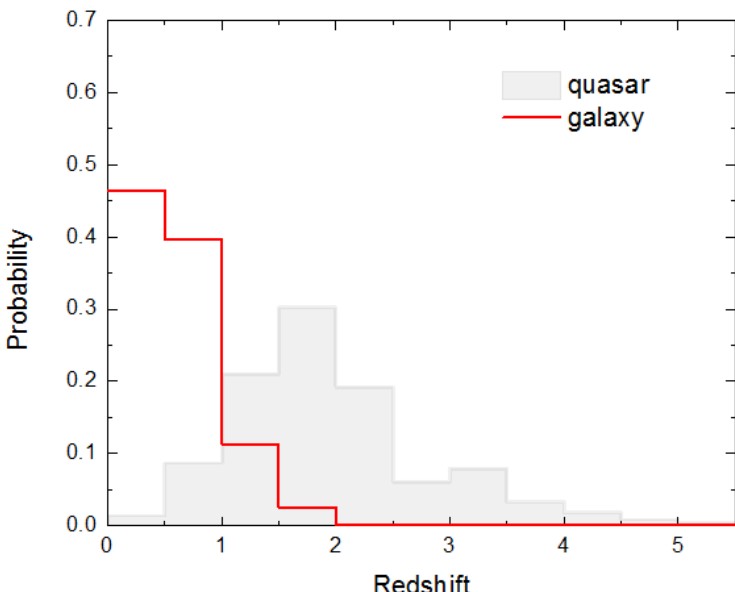

**Figure 1.** Redshift distributions in single lensing events expected to be observed by Roman Space Telescope. The red line shows the redshift distribution of the galaxy and the gray shadow shows the redshift distribution of quasars.

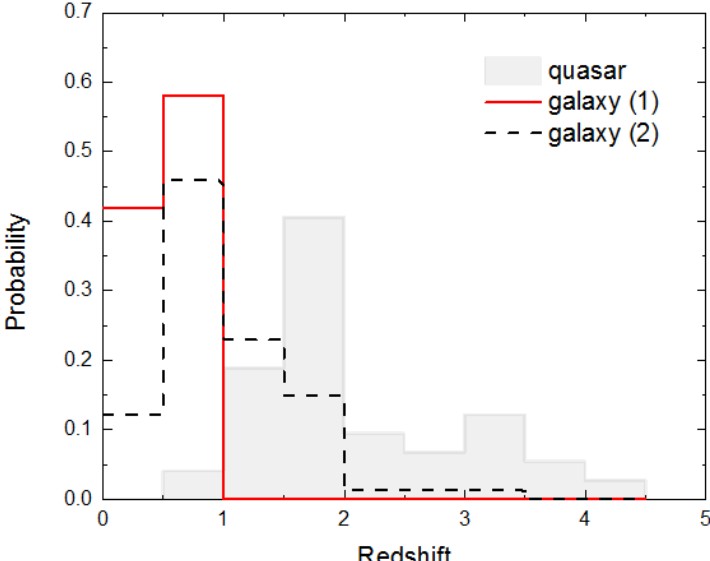

**Figure 2.** Redshift distributions in multiple lensing events expected to be observed by the Roman Space Telescope. The red line shows the redshift distribution of the first galaxy, the dashed black line shows the redshift distribution of the second galaxy and gray shadow shows the redshift distribution of quasars.

The multiple lensing systems can exhibit up to six distinct images. Our findings regarding the distribution of the number of images are visually depicted in Figure 3.

The formation of doubly lensed quasars can occur in lensing systems comprising one or two lenses. Figure 4 displays the distribution of time delays between two images as a function of the Einstein radius. It is evident from Equations (13) and (19) that these two quantities are strongly correlated. The Einstein radii in our study range from approximately 0.11 arcsec (which is the angular accuracy of the Roman Space Telescope) to over 5 arcsec, while the time delays span from around 0.2 h to 2900 days, with a median value of approximately 10 days.

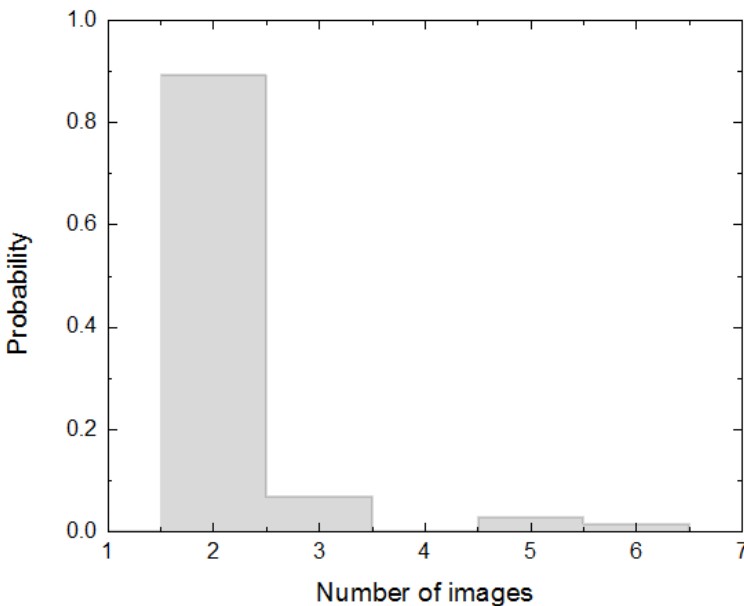

**Figure 3.** Distribution of number of images for strong lensing events caused by multiple lenses. As one can see, up to six images can form (see text for details).

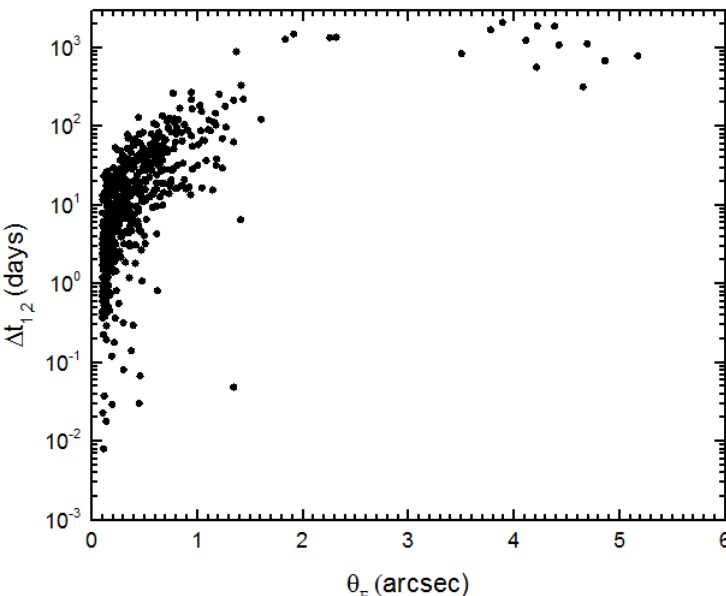

**Figure 4.** Time delay distribution between two lensed images by the Einstein radius.

## 5. Conclusions

In this paper, we estimate the number of quasars that would be observed by the Roman Space Telescope. Our calculation is based on the quasar luminosity function (QLF) obtained from the Spitzer Space Telescope imaging survey [16]. We remark that we used the same QLF for both instruments, even if there is some discrepancy in their band observation. Roman F213 filter will observe (1.95–2.30 µm) (with midpoint in 2.125 µm), whereas the QLF is established for the waveband of 5 µm. We emphasize that the predicted quasar number depends on the sensitivity of the filter. For the deep survey, the number is increased about 20 times (see Table 1).

Moreover, we investigated the case of quasars lensed by one or two foreground galaxies in the case of the Roman Space Telescope. Using the redshift distributions of both quasars and galaxies, together with the galactic mass distribution and velocity dispersion

of galaxies, we find that one quasar among 180 observed by Roman Space Telescope would be lensed by foreground galaxies.

Multiple lensing events are a rather rare phenomenon in astronomy, but it is very likely that the first such events will be discovered soon by the next generation of space telescopes. We find that about 10% of the lensed quasars observable by the Roman Space Telescope are expected to be due to multiple lensing systems. Lensing by more than one galaxy along the line of sight can lead to interesting image configurations. Such systems are expected to be very important, especially in the future, both for constraining lens models of individual systems and for statistical analysis in gravitational lensing.

As discussed in the previous section, if the quasar is time variable, the time delay between the lensed images can be measured. The theoretical expression of the time delay between two lensed images is given by Equation (13). As one can see, it is the product of the time-delay distance $\frac{D_{\Delta t}}{c}$, which depends on cosmology (is inversely proportional to $H_0$), and a term that depends on the lens configuration and its mass distribution. Among these quantities, only the image positions can be defined directly by observations, whereas the lens potential $\Psi(\vec{\theta})$ and the source position $\vec{\beta}$ must be inferred via lens modeling. Although the time delay can provide valuable constraints on the cosmological parameters, there are several limitations and challenges that one would need to take into account: (1) modeling uncertainties (small errors in the modeling of lensing system can lead to significant biases in the inferred cosmological parameters); (2) substructure and environment effects (the presence of substructure within the lensing object and the presence of other nearby galaxies can affect the lensing potential introducing additional time delays or perturbations in the images); (3) systematic errors (sources of systematic errors in strong gravitational lensing include telescope calibration, analysis pipeline, parametric model choice, and assumptions about the lensing object and cosmological model); (4) small sample size (few observed strong gravitational lensing systems with measured time delays limit the statistical power and robustness of the inferred cosmological parameters); (5) degeneracies with other parameters (the Hubble constant measurement from time delay cosmography is affected by degeneracies with respect to other cosmological parameters, which can lead to a broader range of allowed values for $H_0$) [31].

The Roman Space Telescope observations may allow addressing these limitations with its large field of view and high sensitivity, enabling the detection of more strong lensing events and increasing the sample size for improved statistical power. It has superior angular resolution and image quality, allowing for more accurate measurements of the lensing geometry and better identification of substructures. Additionally, operating at near-infrared wavelengths reduces uncertainties caused by dust and other obscuring materials.

We also mention that in the present work, we focused only on the SIS (singular isothermal sphere) model for lensing galaxies. However, we are currently exploring other models and how they impact the estimated time delay between lensed images. The investigation of these additional models and their influence will be addressed in a separate, more comprehensive paper.

**Author Contributions:** Conceptualization, L.H.; M.H. and F.D.P.; methodology, L.H.; M.H. and F.D.P.; software, L.H. and E.G.; validation, L.H., M.H. and F.D.P.; writing—original draft preparation, L.H.; writing—review and editing, F.D.P., M.H. and E.G.; supervision, M.H. and F.D.P. All authors have read and agreed to the published version of the manuscript.

**Funding:** This research received no external funding. The APC was funded by the Faculty of Natural Science, University of Tirana, Albania.

**Institutional Review Board Statement:** Not applicable.

**Informed Consent Statement:** Not applicable.

**Data Availability Statement:** All the results presented in this paper are derived from rigorous numerical calculations based on the equations provided in the paper. No new data was created.

**Acknowledgments:** This article is based upon work from COST Action CA21136—Addressing observational tensions in cosmology with systematics and fundamental physics (CosmoVerse), supported by COST (European Cooperation in Science and Technology). FDP acknowledge the TAsP and Euclid projects of INFN.

**Conflicts of Interest:** The authors declare no conflict of interest.

## Notes

1　　See to https://research.ast.cam.ac.uk/lensedquasars/, accessed on 21 May 2023.

2　　In this work, the lens systems consisting of two lenses are called the multiple lensing systems.

3　　https://roman.gsfc.nasa.gov/science/WFI_technical.html, accessed on 21 May 2023

4　　In the case of a point-mass lens, by solving the Einstein field equations, the deflection angle is $\hat{\alpha} = \frac{4GM}{c^2 r_{min}}$, where $r_{min}$ is the closest approach distance

5　　This definition is only in multiple systems, whereas for its definition in single systems see Equation (4).

6　　It is the sum of many smaller event samples generated separately due to the rather long time necessary for the Monte Carlo simulation to run.

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
