# Peer review of "Investigating Gravitationally Lensed Quasars Observable by Nancy Grace Roman Space Telescope"

_galaxies, doi:10.3390/galaxies11030071_

Round 1

Reviewer 1 Report

The article galaxies-2387399 "Investigating gravitationally lensed quasars observable by Nancy Grace Roman Space Telescope Authors" by Lindita Hamolli, Mimoza Hafizi, Francesco De Paolis, Esmeralda Guliqani presents a well-researched and thoughtful investigation of the potential for strong gravitational lensing to provide insights into various cosmological parameters, with a particular focus on using the Nancy Grace Roman Space Telescope to observe lensed quasars. However, there are a few critical comments to be made:

  1. While the article does an excellent job of discussing the potential benefits of using strong gravitational lensing to test cosmological parameters, it does not sufficiently address the limitations or challenges of this approach. For example, it would be helpful to discuss the various sources of uncertainty that could affect measurements of the Hubble constant using time delays between lensed images, such as the presence of substructure in the lens galaxy or the potential impact of the lensing environment.

  2. The article is very technical and assumes a high level of background knowledge on the part of the reader. While this may be appropriate for a scientific paper, it could make the article less accessible to a wider audience. It might be useful to include more explanatory material or to provide a glossary of technical terms.

  3. The article could benefit from more discussion of the implications of the results presented. For example, the authors mention that detailed studies of lensing systems could allow testing of mass distribution models of galaxies, but they do not elaborate on what such testing might involve or what the potential outcomes could be.

Overall, this is a well-researched and informative article that highlights the potential of strong gravitational lensing to provide insights into cosmological parameters. However, it could be improved by more clearly addressing limitations and challenges, making the material more accessible to a wider audience, and discussing the implications of the results in more detail.

Quality of the English Language is acceptable. 

Round 2

Reviewer 1 Report

The authors have addressed all points raised by me. We believe that the paper has been substantially improved with respect to the previous version.

English is acceptable.